# Antibiotic Use and Resistance Knowledge Assessment of Personnel on Chicken Farms with High Levels of Antimicrobial Resistance: A Cross-Sectional Survey in Ica, Peru

**DOI:** 10.3390/antibiotics11020190

**Published:** 2022-02-01

**Authors:** María Dávalos-Almeyda, Agustín Guerrero, Germán Medina, Alejandra Dávila-Barclay, Guillermo Salvatierra, Maritza Calderón, Robert H. Gilman, Pablo Tsukayama

**Affiliations:** 1School of Veterinary Medicine, Universidad Nacional San Luis Gonzaga, Ica 11004, Peru; maria.davalos@unica.edu.pe (M.D.-A.); agustingcanelo@gmail.com (A.G.); gemegir1@hotmail.com (G.M.); 2Microbial Genomics Laboratory, Department of Cellular and Molecular Sciences, Faculty of Sciences and Philosophy, Universidad Peruana Cayetano Heredia, Lima 15102, Peru; alejandra.davila.b@upch.pe (A.D.-B.); guillermo.salvatierra@upch.pe (G.S.); 3Infectious Diseases Research Laboratories, Department of Cellular and Molecular Sciences, Faculty of Sciences and Philosophy, Universidad Peruana Cayetano Heredia, Lima 15102, Peru; maritza.calderon.s@upch.pe; 4Department of International Health, Bloomberg School of Public Health, Johns Hopkins University, Baltimore, MD 21218, USA; gilmanbob@gmail.com; 5Parasites and Microbes, Wellcome Sanger Institute, Hinxton, Saffron Walden CB10 1RQ, UK

**Keywords:** AMR, public awareness, farmworkers, ESBL, chicken, growth promoters

## Abstract

Poultry farming represents Peru’s primary food animal production industry, where antimicrobial growth promoters are still commonly used, exerting selective pressure on intestinal microbial populations. Consumption and direct animal-to-human transmission have been reported, and farmworkers are at high risk of colonization with resistant bacteria. We conducted a cross-sectional survey among 54 farmworkers to understand their current antimicrobial resistance (AMR) awareness in Ica, Peru. To gain insight into the potential work-related risk of exposure to bacteria, we also measured the AMR rates in *Escherichia coli* isolated among 50 broiler chickens. Farmworkers were unaware of antimicrobial resistance (31.5%) or antibiotic resistance (16.7%) terms. Almost two-thirds (61%) consumed antibiotics during the previous month, and only 42.6% received a prescription from a healthcare professional. A total of 107 *E. coli* chicken isolates were obtained, showing a high frequency of multidrug-resistant (89.7%) and extended-spectrum beta-lactamase (ESBL) production (71.9%). Among ESBL-producer isolates, 84.4% carried the *bla*_CTX-M_ gene. Results identified gaps in knowledge that reflect the need for interventions to increase antimicrobial awareness among poultry farmworkers. The high AMR rates among *E. coli* isolates highlight the need to reduce antimicrobial use in poultry farms. Our findings reveal a critical need for effective policy development and antimicrobial stewardship interventions in poultry production in Ica, Peru.

## 1. Introduction

Peru records one of the largest per capita consumption rates of chicken meat in South America. Poultry farming represents the country’s primary food animal production [1]. The widespread intensive systems of broiler chicken rearing, aimed to meet the high national demand, commonly use antimicrobials as growth promoters [2] to allow for more gut nutrient absorption [3]. The constant exposure to antimicrobials ultimately exerts selective pressure on the chicken’s intestinal microbial populations [4]. In turn, these bacteria select and acquire antimicrobial resistance genes (ARGs) to adapt to their environment.

Consumption of animal products is one of the most common vehicles for introducing resistant *E. coli* strains into human populations [5,6,7]. Direct animal to human transmission of AMR has been reported, and farmworkers are at risk of colonization with resistant bacteria from animals [8,9,10] by various routes, including ingestion, inhalation, and dermal contact [11]. A lack of knowledge and awareness of appropriate antimicrobial use among farm owners and workers may worsen this problem [12,13]. Previous surveys applied to poultry farmworkers [14,15,16,17], drug vendors [18,19], and the general public [20,21,22,23] have exposed a poor understanding of the problem encompassing antibiotic resistance and the misuse of such drugs in livestock systems and human health. The emergence of multidrug-resistant (MDR) bacteria in animal populations and their potential carriage and gene exchange into clinical settings represent an emerging risk to global public health [24].

The World Health Organization’s Global Action Plan on AMR addresses the need to strengthen knowledge and evidence-based practices through surveillance and research [25]. Accordingly, current national strategies designed to tackle the problem from all aspects of One Health have been proposed, starting with food chain surveillance [26]. In many low and middle-income countries (LMIC), antimicrobial use in poultry is not regulated, leading to misuse and facilitating the emergence and spread of AMR [27]. Baseline information and surveillance studies are scarce in Peru compared to other LMICs [28]. However, the potential dissemination of ARGs of gut bacteria from commercial chicken meat to humans with different degrees of exposure has recently been described [29].

Based on the lack of data concerning knowledge and awareness of AMR and antibiotic use among Peruvian poultry farmworkers, we surveyed individuals working in broiler chicken farms in Ica, one of the main poultry producing regions in Peru, to help understand the current state of awareness and common behaviors related to antimicrobial use in the workplace. Additionally, based on their potential work-related exposure to AMR transmission, we aimed to measure AMR rates in *E. coli* isolated from broiler chickens belonging to farms in the same area. We focused on the phenotypic and genotypic determinants of extended-spectrum beta-lactamase (ESBL) production, an important resistance mechanism associated with severe infections in hospital and community settings [30,31,32].

## 2. Results

Farmworker cross-sectional survey: The adapted antimicrobial knowledge and awareness survey was applied to 54 workers from various farms in Chincha Province, Ica, Peru. The mean age was 38.9 (range 21–66 years), and only 5.6% (three in 54) of the participants were women, two of which were veterinarians and one a vaccinator. Most male respondents were farm operators whose activities involved close contact with the feed mill and the birds, either as handlers or vaccinators, including veterinarians. According to their educational level, most participants only had an early or primary education (77.8%). A total of 61% reported taking antibiotics during the last month for personal use, and 68.5% incorrectly agreed with the statement that it was adequate to take antibiotics prescribed for friends or family as long as they were used to treat the same illness. In all, 42.6% said that they used antibiotics only when they received a prescription from a healthcare professional, and 20.4% stopped antibiotic treatment once they felt better (Table 1).

A total of 61.1% (33 in 54) of the participants had taken antibiotics during the previous month (Table 1, Q1), from which all were male. Participants between 34 and 43 years (39.4%, 13 out of 33) and with an early/primary educational level (78.8%, 26 out of 33) presented the highest frequency of antibiotic consumption during the last month. However, no significant differences were found (*p* > 0.05, Fisher’s exact test, see Table 2). One question proposed a list of different illnesses and medical conditions, asking if they could be treated with antibiotics (Table 1, Q7). Among the listed diseases, only skin infection, gonorrhea, and bladder/urinary tract infection should be treated with antibiotics. The majority of respondents (74.1%, *n* = 40) correctly indicated bladder/urinary tract infections as pathologies treatable with antibiotics. Overall, 55.6% correctly selected gonorrhea and only 14.8% skin infections. Several farmworkers were unaware of infectious agents involved in the listed diseases, suggesting a treatment based on antibiotics for diarrhea (77.8%), HIV/AIDS (68.5%), fever (51.9%), measles (40.7%), cold/flu (27.8%), headaches (18.5%), sore throat (18.5%), body aches (14.8%), and malaria (9.3%). Results by educational level and age category are detailed in Figure 1.

A total of 31.5% of participants were unaware of the term antimicrobial resistance and 16.7% of antibiotic resistance. Moreover, only 33.3% had heard about antibiotic-resistant bacteria. The respondents answered eight queries regarding AMR with true or false answers (Table 1, Q9). A total of 88.9% correctly identified the statement that people should use antibiotics only when a doctor or nurse prescribes them. Most farmworkers (92.6%) responded that doctors should only prescribe antibiotics when needed, and 61.1% agreed not to keep antibiotics from one treatment and use them for other later illnesses. Additionally, 42.6% thought that farmers should give fewer antibiotics to food-producing animals. A group of respondents incorrectly agreed that having child vaccinations up to date (38.9%) and washing hands (24.1%) are good ways to help address the problem of antibiotic resistance.

The calculated knowledge score regarding antibiotic use resulted in a mean of 7.3 (SD:2.2) out of 14 points among all participants. The score of participants with a secondary educational level was higher than early/primary school graduates (Figure 2A). The oldest age group (>43 years) had the highest knowledge level on good antibiotic use, followed by the younger participants (<34 years) (Figure 2B). Participants who had taken antibiotics during the previous month showed better knowledge of antibiotics than those who reported not having taken any antibiotics (Figure 2C). However, no significant differences were found for educational level (*p* > 0.05, *t*-test), antibiotic consumption during the last month (*p* > 0.05, *t*-test), and age category (*p* > 0.05, one-way ANOVA).

Determination of antibiotic resistance in commensal *E. coli* from chickens: *Escherichia coli* isolates (*n* = 107) were obtained from cloacal swabs of 50 broiler chickens from three different poultry farms (Farm A = 32, Farm B = 37, Farm C = 38) from three districts in Chincha, Ica. Susceptibility results for all isolates revealed an 89.7% multidrug-resistant (MDR) phenotype, with no statistical difference between the three farms (*p* > 0.05, Fisher’s exact test). High resistance levels were found for trimethoprim/sulfamethoxazole (95.3%), amoxicillin (86.9%), nalidixic acid (85.1%), tetracycline (80.4%), and cefalotin (78.5%). No resistance to meropenem was found among isolates (see Table 3).

Farm B had the highest frequency of resistant isolates to several antimicrobials, including nalidixic acid, amoxicillin, cefalotin, ciprofloxacin, and chloramphenicol. A high percentage of ESBL-producing *E. coli* (71.9%, 77/107) was identified, specifically in Farm B (89.2%, 33/37), compared to the other two (*p* = 0.012, Fisher’s exact test). Moreover, all ESBL-producing isolates were identified as MDR, and 84.4% (65/77) carried *bla*_CTX-M_, with no statistical difference between farms (*p* > 0.05, Fisher’s exact test).

## 3. Discussion

The survey utilized an adaptation of a WHO questionnaire on AMR to investigate antibiotic use practices and knowledge among 54 farmworkers from broiler chicken farms. Several respondents were unaware of which pathologies should be treated with antibiotics and evidenced misconceptions about AMR. Participants had insufficient awareness of antibiotic-related terms, such as antimicrobial resistance or antibiotic-resistant bacteria. Moreover, some participants incorrectly correlated that having child vaccinations up to date and washing hands are good ways to address the problem of antibiotic resistance.

Participants who obtained a better antibiotic knowledge score of antibiotic use had a higher educational level. This finding matches with an observation described in the WHO survey that people with a lower education level are more likely to incorrectly use antibiotics than people with higher educational levels [33]. Older participants (>43 years) showed better antibiotic knowledge than the youngest age group (<34 years). However, younger cohorts often show good knowledge about antibiotics and antibiotic use [34,35]. In rural areas, people dealing with poultry are relatively older people, and their results could be associated with work experience or previous interactions with veterinarians.

Participants who had taken antibiotics during the last month obtained a better antibiotic knowledge score. Apparently, farmworkers with a recent exposure gain sufficient yet not comprehensive knowledge about antibiotics. Similar results have been previously reported among the general population [36,37]. Our findings highlight the need to train farmworkers on AMR as a potential measure to reduce the unregulated use of antimicrobials on farm animals. Actions that effectively build an understanding of how and when to take antimicrobials are critical among farmworkers in poultry settings. Effective interventions and educational programs delivered by health care professionals are needed to train farmworkers to raise awareness about AMR, as those part of the National Multi-sectoral Action Plan to Combat Antimicrobial Resistance, which include workshops and information dissemination on AMR through social media [26].

To gain insight regarding the potential work-related risk of exposure to AMR bacteria through chickens, our study measured, over a year, the rates of antimicrobial resistance in *E. coli* isolated from broiler chickens in a high-producing region in Peru. We found high levels of resistance and MDR phenotypes in most isolates. Our findings showed high resistance rates to antimicrobials commonly used in poultry farms, including trimethoprim/sulfamethoxazole, amoxicillin, nalidixic acid, tetracycline, and cefalotin. The use of antimicrobials in poultry production increases the selective pressure for commensal bacteria such as *E. coli* [4]. The increasing AMR rates in *E. coli* from poultry constitute a significant threat to human and animal health, with animals serving as zoonotic reservoirs of resistant bacteria [38]. Farm B showed the highest rates of ESBL phenotypes at 89.2%. Plasmids that encode ESBLs tend to carry genes giving resistance to antimicrobials such as quinolones, aminoglycosides, and sulfonamides [39]. While the same animal density was reported for the three farms, Farm B had a greater flock size. Pathogens can be introduced to a flock through various routes, including workers, feed, water, fomites, and other animals [40,41,42]. The high frequency of MDR *E. coli* in Farm B might be explained by a larger farm involving more workers, increasing potential contamination routes.

We observed a higher frequency of MDR ESBL-producing *E. coli* in Farm B than the others. Farm B reported using Zinc Bacitracin and Colistin Sulfate as growth promotors. However, we did not test susceptibility against those antibiotics, and we could not establish an association between the growth promotors used and the high levels of MDR isolates found. Based on previous reports, we hypothesize that a high prevalence of resistance in Farm B could be linked to a greater flock size and elevated temperature and humidity levels inside the houses, resulting in heat stress and consequent watery droppings that increase bedding humidity, which facilitates bacterial survival and colonization [43]. However, housing environmental conditions were not measured in this study. Among ESBL-producing isolates, 12 (15.6%) were negative for *bla*_CTX-M_. This may be explained by the presence of other ESBL genes, such as *bla*_TEM_ or *bla*_SHV_ [44].

Dispensing therapeutic or prophylactic antibiotic doses in feed or water for mass administration and flock treatment is common in local and rural farms [45]. All farms reported administering Zinc Bacitracin to the birds during the pilot study period. Moreover, Farm B also administered colistin sulfate in feedstuff. Colistin is considered a last-resort drug for treating severe human clinical infections caused by MDR Gram-negative bacteria [36] and has been widely used in local animal production for decades. Even though a ban on polymyxin E import and trade in the country was established in 2019, local commerce still allows its use until stock depletion [46]. Supplementation with commercially available premixes containing sub-inhibitory amounts of antimicrobials, also a common local practice, is regarded to positively affect growth and aid with feed conversion [47]. However, antibiotics used as growth promoters alter the microbiota and generate a selective pressure that increases the rate of AMR in the microbiota of farm animals [48]. The elevated frequency of MDR *E. coli* isolates obtained in this study highlights the potential consequences of AGPs in poultry production and warrants further investigation of their impact as a feed additive in local settings.

This study had limitations. The cross-sectional survey focused on a small set of questions targeted at general knowledge and antimicrobial drugs usage. It was applied to a limited number of farmworkers in three farms. Future research should include more participants to address the full complexity of antibiotic knowledge and use and expand on questions specific to animal agriculture relating to current practices and beliefs concerning the antibiotic supplementation and treatment of food animals. Notably, efforts should be directed towards understanding the directionality of current practices [17] and the main drivers [49,50,51] of AMR in local poultry systems, as well as quantifying antibiotic use [52]. Temperature and humidity levels inside the poultry houses would have helped explain some of the data more accurately. Peru’s poultry production model usually reuses bedding across the year, which could serve as a vehicle for MDR bacteria and ARGs from previous batches. In that sense, the frequency of bedding change could affect our results and should be considered in future studies, including manure management and its impact on other agricultural systems [53]. Our results may not represent the AMR situation in the poultry industry in Peru. However, they provide evidence of highly resistant *E. coli* in animal production. They should alert veterinary and public health stakeholders to control and limit antibiotic use in poultry production. The data could serve as a baseline for future qualitative AMR risk assessment frameworks [54]. A study of AMR clustering among farmworkers, chickens, and farm environments [55] integrating novel genomic techniques [56] could provide detailed insight into AMR transmission in foodborne pathogens and exposure risks in poultry farms in the region. Although it was not measured, these results hint at the possible work-related risk of exposure to highly resistant bacteria. Due to the nature of our results and considering the scarce publicly available data on AMR in the studied area, the results will be translated to Spanish and shared amongst the local agrarian and environmental health services, farmworkers, and owners.

## 4. Material and Methods

### 4.1. Farmworker Cross-Sectional Survey

Fifty-four workers from broiler chicken farms in Chincha province, northern Ica, Peru, were recruited after consenting to be surveyed on their knowledge and personal use of antibiotics. The World Health Organization (WHO) questionnaire: “Antibiotic resistance: multi-country awareness survey” [33], was translated into Spanish and modified in some sections to accommodate its application (see Appendix A). The survey included nine questions with multiple-choice and true/false responses. It was applied by the face-to-face method, conducted by trained researchers for the answers to remain anonymous, and included demographic information, such as age, gender, and educational level. We generated a knowledge score based on Q5, Q6, and Q7. To calculate the score, all participants had to indicate whether Q5 and Q6 statements were false and correctly specify in Q7 which of the 12 different illnesses and medical conditions should be treated with antibiotics. Thus, we generated a total score of 14 points.

### 4.2. Study Farms

Three broiler chicken farms from the same region in Ica (Appendix A), were included (Figure 3A). The farms share market and biosecurity characteristics of Sector 2 of the FAO/OIE (2007) classification of poultry production systems [57], consisting of intensive semi-technified commercial productions with moderate to high biosecurity levels (Figure 3B). Flock sizes varied: Farm A had approximately 16,000 birds, Farm B 75,000, and Farm C 42,000. Yet, the three farms reported the same animal density of 8.5 chicken/m^2^. All farms are located near roads leading to rural populated areas where other poultry and livestock productions also converge. We recorded information on the health status and antimicrobial drugs supplied to the flocks during the sampling period. All three farms reported zinc-bacitracin use as an antibiotic growth promoter (AGP) during the sampling periods. The use of colistin sulfate was reported in Farm B only.

### 4.3. Chicken Samples

We sampled 35-day old healthy broiler chickens (*n* = 50) from the three farms during April, July, and December 2018. The sampling did not interfere with the way birds were raised. Chickens were randomly selected from each flock every time. Sterile swabs were inserted inside each bird’s cloaca and rotated clockwise, securing contact with the mucosal surface. Cloacal swabs were transported in sterile saline solution tubes at 4 °C within 2 h to the laboratory for bacterial culture.

### 4.4. Bacterial Culture and Antibiotic Susceptibility Testing

Samples were streaked in MacConkey agar (Becton Dickinson, Heidelberg, Germany) and incubated at 37 °C for 24 h. Three presumptive colonies per plate were selected for identification with a biochemical profiling panel, including Simmons Citrate, Triple Sugar Iron Agar, MIO medium (Motility, Indole, Ornithine), Lysine Iron Agar, and Methyl Red Voges-Proskauer Broth (Becton Dickinson). Those confirmed as *Escherichia coli* were included in the study and stored at −20 °C in Tryptic soy broth (TSB, Becton Dickinson) with 10% glycerol. Disk diffusion tests were performed for chloramphenicol (30 µg), meropenem (10 µg), nalidixic acid (30 µg), ciprofloxacin (5 µg), gentamicin (10 µg), azithromycin (15 µg), sulfa-trimethoprim (1.25 µg + 23.75 µg), tetracycline (30 µg), amoxicillin (20 µg), cefalotin (30 µg), and cefepime (30 µg) according to CLSI standards [58], using susceptible, intermediate, and resistant definitions. Extended-spectrum β-lactamase (ESBL) detection was performed using the cefotaxime-ceftazidime-cefepime-aztreonam and amoxicillin with clavulanic acid test [59].

### 4.5. DNA Extraction

We adapted a protocol based on heat treatment followed by boiling to release bacterial DNA [60]. Three to four colonies were picked from each isolate grown in Trypticase Soy Agar plates and diluted 1:4 in 200 µL of Tris-EDTA Buffer solution in 1.5 mL sterile tubes and vortexed. The tubes were then placed in a dry-heat plate at 100 °C for 10 min and centrifuged at 14,000 rpm for 5 min. The supernatant was transferred to a sterile tube for use in PCR assays.

### 4.6. Detection of bla_CTX-M_ Gene

All positive isolates for phenotypic ESBL production were tested for the presence of the *bla*_CTX-M_ gene. The primers used were 5′-TTTGCGATGTGCAGTACCAGTAA-3′ and 5′-CGATATCGTTGGTGGTGCCATA-3′, as previously described [61]. These amplify a conserved 544 bp fragment common to most *bla*_CTX-M_ genes. PCR was carried out in a 25 µL reaction containing the following concentrations: 2 mM MgCl_2_, 150 µM dNTPs, 1 µM of each primer, 1 Unit of Taq polymerase, 1× PCR buffer (10 mM Tris-HCl, 50 mM KCl), and 2 µL DNA template (20–50 ng). Reactions were performed with a 5-min denaturation at 94 °C, 35 annealing cycles at 94 °C, 58 °C, and 72 °C of 30 s each, and a final extension of 5 min at 72 °C on a PTC-150 thermocycler (MJ Research, Inc., Watertown, MA, USA). Amplification products were resolved on a 2% agarose gel with ethidium bromide. As positive controls, we used isolates with at least one of the *bla*_CTX-M_ genes, confirmed by whole-genome sequencing (WGS) from a previous study [29].

### 4.7. Data Analysis

A bivariate analysis to compare antibiotic consumption during the last month between gender, educational level, and age category was performed using Fisher’s exact test. The calculated knowledge score about antibiotics was compared for educational level and age category using a *t*-test and one-way analysis of variance (ANOVA), respectively. The Clinical & Laboratory Standards Institute (CLSI) guidelines were used to categorize isolates as susceptible, resistant, or intermediate. Multidrug resistance (MDR) was defined as an isolate expressing phenotypic resistance to three or more antibiotics classes [62]. A bivariate analysis to compare resistance results between sampled farms was performed using Fisher’s exact test. Statistical analysis was performed with a 95% confidence level using STATA 16 (Stata Corp., College Station, TX, USA).

## 5. Conclusions

This study’s survey results indicate insufficient knowledge amongst farmworkers regarding the antimicrobial resistance problem and the appropriate and prudent use of antimicrobial drugs for treating human diseases. *E. coli* isolates from chickens raised for human consumption showed high resistance rates to various antimicrobials used in human clinical settings. Our results highlight the need to (1) promote antibiotic knowledge and awareness among farmworkers, (2) implement measures to reduce the use of antimicrobials in poultry systems in Peru, and (3) establish surveillance systems to monitor the rates of antimicrobial-resistant bacteria in local chicken populations.

## Figures and Tables

**Figure 1 antibiotics-11-00190-f001:**
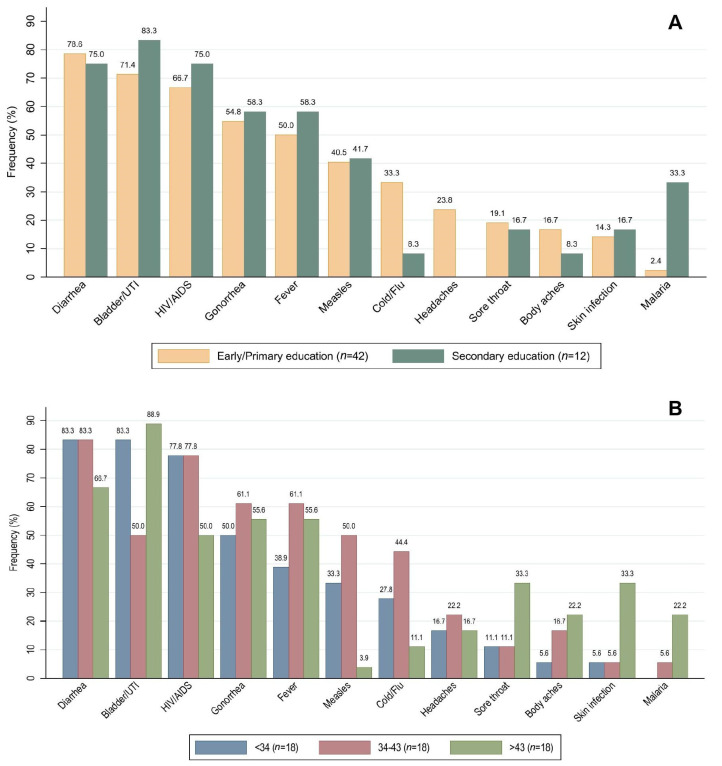
Frequency of responses to question: “Do you think these conditions can be treated with antibiotics?” Medical conditions to be treated with antibiotics: Gonorrhea, Bladder/UTI and Skin infection. (**A**) results by educational level, (**B**) results by age category in years.

**Figure 2 antibiotics-11-00190-f002:**
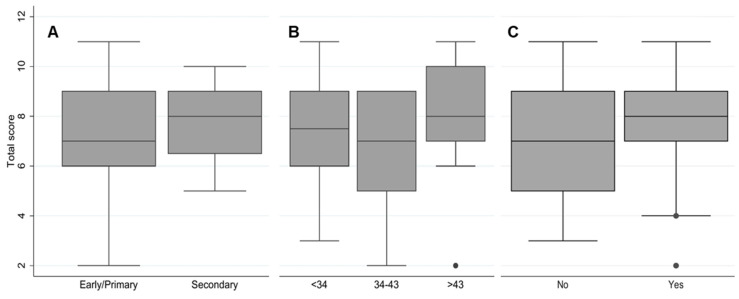
Calculated knowledge score regarding antibiotic use among participants. (**A**) educational level (*p* = 0.420, *t*-test), (**B**) age category in years (*p* = 0.276, one-way ANOVA), (**C**) antibiotic consumption during the last month (*p* = 0.432, *t*-test).

**Figure 3 antibiotics-11-00190-f003:**
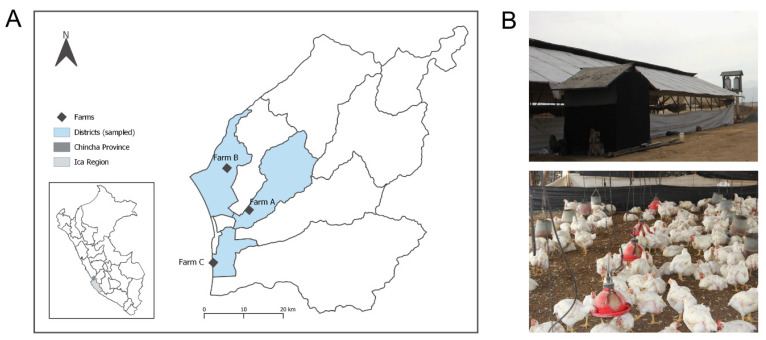
(**A**) Geographical location of the sampled farms along Chincha Province in Ica region, Peru. (**B**) Exterior and interior views of the chicken farms. The provincial map of Ica was created using QGIS v3.16.0 (https://qgis.org) (accessed on 23 February 2021).

**Table 1 antibiotics-11-00190-t001:** Cross-sectional survey results of a sample of 54 farm workers from Ica poultry farms.

Results		*n* (%)
Q1. When did you last take antibiotics?		
In the last month		33 (61.0)
In the last 6 months		19 (35.2)
In the last year		1 (1.9)
More than a year ago		1 (1.9)
Q2. On that occasion, did you get the antibiotics (or a prescription for them) from a doctor or nurse?		
Yes		23 (42.6)
Q3. On that occasion, where did you get the antibiotics?		
Medical store or pharmacy		36 (66.7)
I had them saved up from a previous time		18 (33.3)
Q4. When do you think you should stop taking antibiotics once you’ve begun treatment?		
When you feel better		11 (20.4)
When you’ve taken all of the antibiotics as directed		42 (77.8)
Don’t know		1 (1.9)
Q5. “It’s okay to use antibiotics that were given to a friend or family member, as long as they were used to treat the same illness” (TRUE)		
Yes		37 (68.5)
Q6. “It’s okay to buy the same antibiotics, or request these from a doctor, if you’re sick and they helped you get better when you had the same symptoms before” (TRUE)		
Yes		25 (46.3)
Q7. Do you think these conditions can be treated with antibiotics?		
Diarrhoea		42 (77.8)
Bladder infection or urinary tract infection		40 (74.1)
HIV/AIDS		37 (68.5)
Gonorrhoea		30 (55.6)
Fever		28 (51.9)
Measles		22 (40.7)
Cold and flu		15 (27.8)
Sore throat		10 (18.5)
Headaches		10 (18.5)
Skin or wound infection		8 (14.8)
Body aches		8 (14.8)
Malaria		5 (9.3)
Q8. Have you ever heard of any of the following terms?		
Antibiotic resistance		45 (83.3)
Superbugs		11 (20.4)
Antimicrobial resistance		37 (68.5)
AMR		5 (9.3)
Drug resistance		38 (70.4)
Antibiotic-resistant bacteria		18 (33.3)
Q9. Do you agree that the following actions would help address the problem of antibiotic resistance? (Yes)		
People should use antibiotics only when they are prescribed by a doctor or nurse		48 (88.9)
Farmers should give fewer antibiotics to food-producing animals		23 (42.6)
People should not keep antibiotics and use them later for other illnesses		33 (61.1)
Parents should make sure all of their children’s vaccinations are up-to-date		21 (38.9)
People should wash their hands regularly		13 (24.1)
Doctors should only prescribe antibiotics when they are needed		50 (92.6)
Governments should reward the development of new antibiotics		18 (33.3)
Pharmaceutical companies should develop new antibiotics		22 (40.7)

**Table 2 antibiotics-11-00190-t002:** Participant’s characteristics and use of antibiotics within the previous month.

Characteristics	Total	Antibiotics Consumed during the Previous Month		*p*-Value *
		Yes (*n* = 33)	No (*n* = 21)	
Age (tertiles)				
<34	18 (33.3)	8 (24.2)	10 (47.6)	0.296
34–43	18 (33.3)	13 (39.4)	5 (23.8)	
>43	18 (33.3)	12 (36.4)	6 (28.6)	
Education level				
Early/Primary	42 (77.8)	26 (78.8)	16 (76.2)	1.000
Secondary	12 (22.2)	7 (21.2)	5 (23.8)	

* Fisher exact test, 95% confidence level.

**Table 3 antibiotics-11-00190-t003:** Antimicrobial resistance rates of *E. coli* from chickens among sampled farms.

Results	Total (*n* = 107)	Farm A (*n* = 32)	Farm B (*n* = 37)	Farm C (*n* = 38)	*p*-Value *
MDR					
Yes	96 (89.7)	28 (87.5)	36 (97.3)	32 (94.2)	0.147
ESBL					
Yes	77 (71.9)	20 (62.5)	33 (89.2)	24 (63.2)	0.012
Amphenicols					
Chloramphenicol	72 (67.3)	22 (68.8)	21 (56.8)	29 (76.3)	0.200
Tetracyclines					
Tetracycline	86 (80.4)	27 (84.4)	23 (62.2)	36 (94.7)	0.002
Sulfonamides					
Trimethoprim/sulfamethoxazole	102 (95.3)	31 (96.9)	37 (100.0)	34 (89.5)	0.078
Aminoglycosides					
Gentamicin	64 (59.8)	23 (71.9)	21 (56.8)	20 (52.6)	0.246
Macrolides					
Azithromycin	2 (1.9)	0 (0.0)	1 (2.7)	1 (2.6)	1.000
Penicillins					
Amoxicillin	93 (86.9)	29 (90.6)	35 (94.6)	29 (76.3)	0.067
Cephalosporins					
Cefalotin	84 (78.5)	21 (65.6)	35 (94.6)	28 (73.7)	0.007
Cefepime	12 (11.2)	4 (12.5)	5 (13.5)	3 (7.9)	0.742
Carbapenems					
Meropenem	0 (0.0)	0 (0.0)	0 (0.0)	0 (0.0)	N.A.
Quinolones					
Nalidixic Acid	91 (85.1)	26 (81.3)	36 (97.3)	29 (76.3)	0.021
Ciprofloxacin	72 (67.3)	15 (46.9)	32 (86.5)	25 (65.8)	0.002

* Fisher exact test, 95% confidence level. N.A.: Not applicable.

## Data Availability

Non applicable.

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
