# Peer review of "Antibiotic Use and Resistance Knowledge Assessment of Personnel on Chicken Farms with High Levels of Antimicrobial Resistance: A Cross-Sectional Survey in Ica, Peru"

_antibiotics, 2022, doi:10.3390/antibiotics11020190_

Round 1
Reviewer 1 Report
The submitted manuscript “Antibiotic use and resistance knowledge assessment of personnel in chicken farms with high levels resistance: a cross-sectional survey in Ica, Peru” focus on a very important problem to human health.
The Tittle of the manuscript should be improved to reflect the specific content of the study and to be more attractive.
The methodology is appropriate, but could be improved. Information about the farm’s structures (like equipment exchange places for workers, other animal species -cats, dogs- in the farms) should be provided since can be a way of contamination between animals and workers.
The results are correctly exposed. The discussion should be improved. The fact that Farm B uses 2 antibiotics as growth promoters can be related to the high frequency of multidrug-resistant and extended-spectrum beta-lactamase (ESBL) production in that farm. This issue should be highlighted in the discussion.
In general, the study brings consistent information to the scientific community and can be a form to alert the local authorities to this problem. However, farmworkers do not read this type of publications and local measures should be promoted to give information and formation to farmworkers and all population. For instance, training actions in each farm, to teach in primary schools, etc can improve the information of the population.
Author Response
The submitted manuscript "Antibiotic use and resistance knowledge assessment of personnel in chicken farms with high levels resistance: a cross-sectional survey in Ica, Peru" focus on a very important problem to human health.
- The Tittle of the manuscript should be improved to reflect the specific content of the study and to be more attractive.
The manuscript's title was reviewed as the reviewer suggested. We believe that its wording reflects the contents of the study as accurately as possible and follows a similar style as previous publications on these topics.
- The methodology is appropriate, but could be improved. Information about the farm's structures (like equipment exchange places for workers, other animal species -cats, dogs- in the farms) should be provided since can be a way of contamination between animals and workers.
We used the FAO/OIE (2007) classification of poultry production systems to describe the farms uniformly. In this case, all the three farms shared market and biosecurity characteristics of Sector 2 of this classification. We then described flocks sizes/density in more detail and the level of biosecurity (moderate to high) which, to the industry's standard, does not allow for other animal species to be present on the farm. We did not see any animals during sample collection, which was not included in the description.
- The results are correctly exposed. The discussion should be improved. The fact that Farm B uses 2 antibiotics as growth promoters can be related to the high frequency of multidrug-resistant and extended-spectrum beta-lactamase (ESBL) production in that farm. This issue should be highlighted in the discussion.
That is correct. Farm B reported using Zinc Bacitracin and Colistin Sulfate as growth promotors. However, we did not test susceptibility against those antibiotics. With our data, we can not establish an association between the growth promotors used and the high levels of MDR and ESBL isolates found in farm B. In the manuscript, we described our hypothesis to the potential reason behind the high levels of MDR and ESBL profiles in contrast to the other two farms: a higher flock size involving more workers and increasing potential contamination routes.
- In general, the study brings consistent information to the scientific community and can be a form to alert the local authorities to this problem. However, farmworkers do not read this type of publications and local measures should be promoted to give information and formation to farmworkers and all population. For instance, training actions in each farm, to teach in primary schools, etc can improve the information of the population.
Once this study is published, we plan to translate it into Spanish and distribute it among local authorities. This measure will make this information more accessible to the general population.
Reviewer 2 Report
First of all, well done. This is a simple, yet effective study, that highlights an important issue. It is very well written; (mostly) easy to read, without lacking information.
My comments are below, they are mostly minor, apart from comments on the Results section - which needs work.
GENERAL:
- The very first sentence "Peru has one of the largest per capita consumption of chicken meat in South America" this part isn't grammatically correct and sounds strange. Please re-word.
- Table 1. There are some alignment issue there, which I presume typesetting will correct.
- It is unclear how the 'knowledge score' was calculated. Please give more information. From the text I cannot tell the questions and calculations that went into deriving that score.
- Methods says there are 10 questions, but I only see 9 in Table 1. Maybe the 10th is there in the text, but the fact this is not clear is an issue.
- Line 71-83. The second half of this paragraph needs re-writing. It isn't as clear as it could be.
- I like that the discussion acknowledges study limitations, not enough people do this.
- Survey work really benefits from the publication of the raw data. I would highly recommend publishing the raw data (at respondent level). It is not only important for science and policy, but could boost your citations significantly. I myself conduct survey work in a similar field, I recently found a paper in my field with raw data and it is going to receive multiple from me this year, purely on that basis.
- Not essential: Growth promoters, especially antibiotic growth promoters, are extremely controversial. Indeed, they are banned in many countries. It may be interesting for readers to hear the authors views on that, in this context. For example, this study highlights how a lack of education can exacerbate the problems with antibiotic growth promoters. This undoubtedly reduces their sustainability and impacts the socio-politico-environmental future of them.
RESULTS:
Basically, the survey part of the results section is messy, inconsistent, and difficult to read. Not only is this frustrating, but it potentially hides a lot of information. I would highly recommend going back to the drawing board on how this is presented. Whilst graphs are always nice, it may be a simple table with overall answers in one column, followed by answers for different cohorts (age, education etc).
- I like Table 1. But then after that, the presentation of the survey results is quite messy. Table 1. gives respondent answers for questions 1-9. This is then followed by Figure. 2, which gives more information on Q9. Then there is Figure. 3, which jumps all the way back to Q1.
- You break down Q9 into Education and Age. You break down Q1 into Education, Age, and Gender. You break down 'Knowledge Score' into Education, Age, and Antibiotic Use. This is inconsistent in multiple ways (1) You present data differently each time (2) You use different groupings each time (3) Why is this just for those parts and not all questions? This presentation is inconsistent and somewhat strange.
Author Response
- The very first sentence "Peru has one of the largest per capita consumption of chicken meat in South America" this part isn't grammatically correct and sounds strange. Please re-word.
The first sentence of the Introduction section was edited according to the reviewer's suggestion:
From: "Peru has one of the largest per capita consumption of chicken meat in South America, and poultry farming represents the country's primary food animal production."
To: "Peru records one of the largest per capita consumption of chicken meat in South America. Poultry farming represents the country's primary food animal production."
- It is unclear how the 'knowledge score' was calculated. Please give more information. From the text I cannot tell the questions and calculations that went into deriving that score.
A more detailed description of knowledge score calculation was included in the text.
- Methods says there are 10 questions, but I only see 9 in Table 1. Maybe the 10th is there in the text, but the fact this is not clear is an issue.
We have corrected the number of questions in the text. As is detailed in Table 1, we used only nine questions.
- Line 71-83. The second half of this paragraph needs re-writing. It isn't as clear as it could be.
We have re-wrote and clarified the sentences.
- Survey work really benefits from the publication of the raw data. I would highly recommend publishing the raw data (at respondent level). It is not only important for science and policy, but could boost your citations significantly. I myself conduct survey work in a similar field, I recently found a paper in my field with raw data and it is going to receive multiple from me this year, purely on that basis.
We have discussed this with all co-authors, and we will not publish the raw data at a respondent level with the manuscript. However, we will share the complete data with everyone that contact the corresponding author.
- Not essential: Growth promoters, especially antibiotic growth promoters, are extremely controversial. Indeed, they are banned in many countries. It may be interesting for readers to hear the authors views on that, in this context. For example, this study highlights how a lack of education can exacerbate the problems with antibiotic growth promoters. This undoubtedly reduces their sustainability and impacts the socio-politico-environmental future of them.
That is correct. Supplementation with commercially available premixes is a common local practice and positively affect growth and aid with feed conversion. However, antibiotics used as growth promoters alter the microbiota and generate a selective pressure that increases the rate of AMR in the microbiota of farm animals. We have mentioned the pros and cons of growth promoters use in the discussion section. Moreover, we had previously included a line in the last paragraph of the discussion about our results potentially serving to alert veterinary and public health stakeholders to control and limit antibiotic use in poultry production.
- Basically, the survey part of the results section is messy, inconsistent, and difficult to read. Not only is this frustrating, but it potentially hides a lot of information. I would highly recommend going back to the drawing board on how this is presented. Whilst graphs are always nice, it may be a simple table with overall answers in one column, followed by answers for different cohorts (age, education etc). I like Table 1. But then after that, the presentation of the survey results is quite messy. Table 1. gives respondent answers for questions 1-9. This is then followed by Figure. 2, which gives more information on Q9. Then there is Figure. 3, which jumps all the way back to Q1.
That is correct. Table 1 explains Q1-Q9 results, and Figure 2 information about Q9. Figure 3 uses Q1 but gives information about the obtained knowledge score, calculated using Q5, Q6, and Q7. The information mentioned about Q1 is detailed in Table 2. We have reorganized the 'material and methods' section at the end, and the figures/tables order have changed:
Table 1: Answers to Q1-Q9
Table 2: Information of participant's characteristics with Q1 (antibiotic consumption)
Figure 1: Information about Q9 (illnesses details)
Figure 2: Boxplots about knowledge score differences among groups
Figure 3: Farms location and description
- You break down Q9 into Education and Age. You break down Q1 into Education, Age, and Gender. You break down 'Knowledge Score' into Education, Age, and Antibiotic Use. This is inconsistent in multiple ways (1) You present data differently each time (2) You use different groupings each time (3) Why is this just for those parts and not all questions? This presentation is inconsistent and somewhat strange.
Among the 54 participants, only three were women. Having an unbalanced representation of both genders, we considered not including them in groups comparison and graphics. To avoid this inconsistency, we have also excluded gender in Table 2 for 'knowledge score' breakdown:
Figure 1. Q9 by educational level and age group.
Table 2. Q1 by educational level and age group.
Figure 2. Calculated knowledge score by educational level, age, and antibiotic use.
Reviewer 3 Report
This research could be interesting for the Journal's public since such a threat could be present anywhere. The importance of transmitted resistance is indubitable and the deep source of this has to clearly p=depicted.
Hard points:
- A good regional presentation to gain insight into the potential work-related risk of exposure to bacteria present in chicken farms
- E. coli chicken isolates obtained, showed a 24 high frequency of multidrug-resistant and an extended-spectrum beta-lactamase (ESBL) production (with the high presence of blaCTX-M gene).
- the work rises the critical and rapid need for an effective policy development and antimicrobial stewardship interventions in poultry production in Peru.
- The Cross-sectional survey is a reliable tool in such studies.
What to improve:
- please develop the part of concrete measures envisaged in Peru to improve "interventions to increase antimicrobial awareness among poultry farmworkers"
- please include a figure with the found sequences
- since you used ANOVA, please present the p values and comment the statistical significance for the obtained values
- please present this as a figure containing a legend.
Conclusion: Minor correction
Author Response
This research could be interesting for the Journal's public since such a threat could be present anywhere. The importance of transmitted resistance is indubitable and the deep source of this has to clearly p=depicted.
Hard points:
- A good regional presentation to gain insight into the potential work-related risk of exposure to bacteria present in chicken farms
- coli chicken isolates obtained, showed a 24 high frequency of multidrug-resistant and an extended-spectrum beta-lactamase (ESBL) production (with the high presence of blaCTX-M gene).
- the work rises the critical and rapid need for an effective policy development and antimicrobial stewardship interventions in poultry production in Peru.
- The Cross-sectional survey is a reliable tool in such studies.
What to improve:
- Please develop the part of concrete measures envisaged in Peru to improve "interventions to increase antimicrobial awareness among poultry farmworkers"
Peru's National Multi-sectoral Action Plan to Combat Antimicrobial Resistance (2019-2021) details its first aim to raise "Improve awareness and understanding regarding resistance to antimicrobials (AMR) through effective communication, education and training". Specifically, under its sub-aim "Raising awareness of specific subgroups of the population over AMR", it includes "Awareness of the producers of slaughter animals and aquaculture, as well as food processors of animal origin and vegetable for human consumption and processors of feed". It intends to share relevant information via official government and social media channels and host workshops with farmworkers. We included the National Action Plan information more explicitly in the discussion section.
- Please include a figure with the found sequences
We did not employ sequences for the methodology. Instead, we used DNA of E. coli isolates derived from broiler chickens from a published study (Murray et al., 2021), which had been confirmed to contain one of the blaCTX-M genes, as positive controls for the PCR assays. The primers used for blaCTX-M were universal primers designed by Edelstein, et al (2003). Here is the bioproject containing the sequences used: https://www.ncbi.nlm.nih.gov/bioproject/PRJNA633873/ The link can also be found at the end in Murray et al. (2021) 's article.
- Since you used ANOVA, please present the p values and comment the statistical significance for the obtained values.
ANOVA test was used for knowledge score comparison among age groups only. No statistical difference was found (p>0.05). This result is mentioned in the results section as "However, no significant differences were found for educational level (p>0.05, T-test), antibiotic consumption during the last month (p>0.05, T-test), and age category (p>0.05, one-way ANOVA)."
- Please present this as a figure containing a legend.
The exact p-values obtained in t-tests and the one-way ANOVA test were added in the Figure 2 header.
.